# High-Precision Automatic Calibration Modeling of Point Light Source Tracking Systems

**DOI:** 10.3390/s21072270

**Published:** 2021-03-24

**Authors:** Ruijin Li, Liming Zhang, Xianhua Wang, Weiwei Xu, Xin Li, Jiawei Li, Chunhui Hu

**Affiliations:** 1Key Laboratory of Optical Calibration and Characterization, Anhui Institute of Optics and Fine Mechanics, Hefei Institutes of Physical Science, Chinese Academy of Sciences, Hefei 230031, China; ruijinli@mail.ustc.edu.cn (R.L.); xhwang@aiofm.ac.cn (X.W.); weilxu@aiofm.ac.cn (W.X.); lixin110@aiofm.ac.cn (X.L.); jiawei19@mail.ustc.edu.cn (J.L.); sariell@mail.ustc.edu.cn (C.H.); 2Science Island Branch, Graduate School, University of Science and Technology of China, Hefei 230026, China

**Keywords:** radiometric calibration, modeling, geometric error, high-precision calibration

## Abstract

To realize high-precision and high-frequency unattended site calibration and detection of satellites, automatic direction adjustment must be implemented in mirror arrays. This paper proposes a high-precision automatic calibration model based on a novel point light source tracking system for mirror arrays. A camera automatically observes the solar vector, and an observation equation coupling the image space and local coordinate systems is established. High-precision calibration of the system is realized through geometric error calculation of multipoint observation data. Moreover, model error analysis and solar tracking verification experiments are conducted. The standard deviations of the pitch angle and azimuth angle errors are 0.0176° and 0.0305°, respectively. The root mean square errors of the image centroid contrast are 2.0995 and 0.8689 pixels along the *x*- and *y*-axes, respectively. The corresponding pixel angular resolution errors are 0.0377° and 0.0144°, and the comprehensive angle resolution error is 0.0403°. The calculated model values are consistent with the measured data, validating the model. The proposed point light source tracking system can satisfy the requirements of high-resolution, high-precision, high-frequency on-orbit satellite radiometric calibration and modulation transfer function detection.

## 1. Introduction

With the rapid development of remote-sensing technology, China’s satellite remote-sensing technology can realize global and multisatellite network observations, thereby enabling comprehensive global observation with three-dimensional and high-, medium-, and low-resolution imaging, which has gradually penetrated all aspects of the national economy, social life, and national security [1]. Radiometric calibration is the process of establishing the functional response relationship between the absolute value of the radiance at the entrance pupil of the remote sensor and the digital number of the output image of the remote sensor and determining the radiometric calibration coefficient of the remote sensor data [2,3]. With the development of global remote-sensing quantitative applications, it has become increasingly urgent to improve the level of quantitation in remote-sensing applications of satellite data. On-orbit radiometric calibration and modulation transfer function (MTF) detection by satellite remote sensors are the basis of satellite remote-sensing quantitative applications. Therefore, higher requirements are put forward for the accuracy of remote sensor radiometric calibration and MTF detection [4,5,6,7]. Vicarious calibration, which is not affected by the space environment or satellite state, can account for atmospheric transmission and environmental impacts. This approach, which can help facilitate authenticity and model accuracy tests of on-orbit remote sensors, has been developed rapidly [8]. As a kind of high-spatial resolution satellite site for vicarious calibration equipment, point light sources are light-weight and small and exhibit excellent optical characteristics. Their layout is flexible, and they can be moved easily. The aperture of the convex mirror depends on the pointing accuracy of the system. To ensure reliability, it is desirable to increase the pointing accuracy, reduce the aperture size, and reduce the volume and weight of the point light source. Furthermore, it is desirable to change the number of mirrors to realize on-orbit radiometric calibration and MTF detection of point light sources with different energy levels [9,10]. Point light source radiometric calibration mainly uses the point light source equipment to reflect sunlight into the entrance pupil of the satellite. Upon calculating the equivalent entrance pupil radiance of the satellite combined with the target response value of the remote-sensing image, the calibration coefficient is calculated according to the remote sensor calibration equation. Because this procedure simplifies the radiative transfer process, it has been widely used [11,12,13,14,15].

According to literature research, so far, few countries have carried out on-orbit radiation calibration and MTF detection of point light sources. The United States was the first to carry out this work, followed by France and China. France has adopted active point light source equipment, mainly using high-energy spotlight for on-orbit MTF detection of SPOT5 [16]. The United States and China mainly use reflective point light source equipment to carry out the corresponding experiments [17,18,19,20,21,22,23]. The key to high-resolution satellite on-orbit radiation calibration based on point light sources is to control the direction of the central optical axis of the point light source reflector. When the central optical axis of the reflector points to the sun, the sunlight enters the convex mirror perpendicularly, the reflected light spot is in a divergent state, and the direction points toward the sun. When the central optical axis of the reflector points toward the position of the bisector of the angle between the satellite and the sun, the reflected light spot is reflected toward the satellite direction in a divergent state. If the pointing position of the optical axis at the edge is reflected toward the direction of the satellite due to low pointing accuracy, the satellite may not observe the point light source or may observe only part of the reflected light spot, which may cause the radiation calibration to fail. Therefore, the success or failure of the point light source on-orbit experiment depends on the pointing accuracy, and the pointing accuracy depends on the tracking accuracy of the system. To improve the pointing accuracy of the system, it is necessary to improve the tracking accuracy of the system. The pointing accuracy of the reflector equipment used by American researchers Schiller et al. [24] to implement the SPARC method (specular array radiometric calibration) of radiation calibration is better than ± 0.5°. In particular, a large convex mirror is used to compensate for the lack of pointing accuracy to ensure that the reflection spot enters the pupil of the satellite. However, the processing accuracy of large convex mirrors is difficult to ensure, and this approach is not convenient for engineering practice and application promotion. In China, the Anhui Institute of Optics and Fine Mechanics, Chinese Academy of Sciences, successively conducted on-orbit radiometric calibration experiments and MTF detection based on point light sources [7,12,13,22]. Initially, a large plane mirror was used as the reflection point light source to perform experiments involving medium- and high-orbit satellites on orbit [22,23]. At present, we mainly carry out on-orbit experiments of point light sources based on convex mirrors. Compared with existing foreign point light source systems, the difference is that we use a smaller convex mirror to overcome the disadvantages associated with larger convex mirrors. The advantage of this approach is that it is easy to change the number of mirrors to produce different energy levels of reflected light, which is suitable for different resolutions in satellite radiometric calibration and MTF detection [13]. However, the disadvantage is that the reflection spot decreases due to the reduction of the aperture of the convex mirror, which increases the difficulty of the satellite reliably receiving the reflected spot. Therefore, to ensure that the reflected light spot is reliably incident on the entrance pupil of the satellite, the key technological improvement that needs to be addressed when using a smaller convex mirror is improving the pointing accuracy. Therefore, to improve the pointing accuracy of the system, a high-precision calibration modeling method for a point light turntable based on a solar vector was established [9]. Compared with previous-generation equipment [22], the integrated pointing accuracy of the system could be enhanced; however, a camera with an automatic observation ability was not introduced in the modeling process. Consequently, the system cannot realize automatic calibration, and it is difficult to realize the high-precision calibration of large-scale automatic cooperative work. To realize automatic calibration, the literature [10] proposed a mirror normal calibration method based on the centroid of the solar image; however, in the initial stage of the model, the influencing factors such as equipment placement errors and camera distortion corrections are not considered. Consequently, the calibration accuracy is affected by single-point calibration and the solar image, and the calibration accuracy needs to be further increased.

The abovementioned calibration techniques based on convex mirrors can achieve satisfactory results in radiometric calibration and MTF detection; however, such approaches cannot meet the requirements of high precision, high frequency and use of existing high-resolution satellites. Nevertheless, unattended multipoint automatic and high-precision pointing adjustment technology can satisfy these requirements. Therefore, in this study, based on the development of a point light source turntable tracking system, an automatic calibration modeling method is developed. Moreover, a high-precision automatic geometric calibration model is established. The system can realize network-based remote control, achieve high-precision pointing of the point light source array tracking system, and realize high-frequency and high-efficiency orbit radiation calibration and MTF detection of high-spatial resolution satellites.

The tracking accuracy described in this paper is the basic guaranteed accuracy required to achieve a comprehensive system design accuracy better than 0.1°; therefore, the design accuracy of our system needs to be better than 0.1°. To realize automatic calibration of the point light source array and achieve the purpose of high-precision tracking of the point light source system, this paper focuses on the establishment of a high-precision calibration model of the point light source system. Starting from the composition of the point light source system, the establishment of a coordinate system and the principle of geometric calibration modeling, this paper studies the establishment of a simplified calibration model of the point light source system. On the basis of the simplified calibration model, considering the geometric error parameters and camera lens distortion parameters that affect the tracking accuracy of the system, the automatic high-precision geometric calibration model is further established. Based on the theoretical verification and solution of the model, the inverse solution algorithm of the calibration model is proposed for experimental verification of the calibrated model. Finally, the experimental verification and system tracking accuracy analysis are carried out.

## 2. Principle of Geometric Calibration Modeling

### 2.1. Equipment System Composition and Coordinate System Establishment

The turntable system of the point light source is composed mainly of a posture control module, mirror assembly, camera and electric control system. The posture control module includes a pitching component and an azimuth component. The pitching component adopts a “U”-shaped forked arm structure. The pitch motor drives a pitching turbine through a two-stage reduction device to drive a mirror to rotate around the pitch axis. The azimuth component is driven by an azimuth motor through the two-stage reduction mechanism to cause the rotary table to rotate around the azimuth axis. The reduction ratio of the second reduction device is 1:360. The pitch and azimuth terminals of the equipment are equipped with an encoder detection device to feed back the rotation angle of the rotary table terminal. The detection accuracy of the encoder is 0.02°. The mirror assembly is arranged between the “U”-shaped forked arms to form a pitching rotation axis. The camera is fixed to the top of the mirror assembly to maintain the camera plane parallel to the mirror plane. The field of view is 23° × 17°. The image resolution is 1280 × 1024 pixels. The resolutions of the azimuth and pitch pixel angles are 0.018° and 0.0166°, respectively. The electric control system is arranged at the base and two fork arms. The abovementioned components compose a point light turntable system, as shown in Figure 1a.

To conveniently describe the coordinate position of the sun and a satellite observed from a certain point on Earth’s surface, a coordinate system is established based on the position of the point light source on Earth’s surface. This system is named the northeast upper coordinate system, which is expressed as loc and described as $\left[\begin{array}{ccc} E & N & U_{p} \end{array}\right]$. *E* points due east in the positive direction. *N* points due north in the positive direction. *U*_p_ points in the vertical upward direction against the geocenter in the positive direction. The mirror coordinate system is fixed on the turntable. The right-hand rectangular coordinate system is composed of the z-axis of the central light axis of the mirror, which is described as $\left[\begin{array}{ccc} x_{\mathrm{mir}} & y_{\mathrm{mir}} & z_{\mathrm{mir}} \end{array}\right]$. In addition, $x_{\mathrm{mir}}$ is based on the pitch axis of the turntable and points to the east, and $y_{\mathrm{mir}}$ takes the azimuth axis of the turntable as the baseline, which is consistent with the *U*_p_ direction, with $z_{\mathrm{mir}}$ pointing to the north. The camera coordinate system is established in accordance with the mirror coordinate system, which is described as $\left[\begin{array}{ccc} x_{\mathrm{cam}} & y_{\mathrm{cam}} & z_{\mathrm{cam}} \end{array}\right]$. The establishment of the coordinate system is shown in Figure 1b.

### 2.2. Principle of Geometric calibration Modeling

Based on the principle of central projection and perspective transformation [25,26], in the same coordinate system, a collinear condition equation is established using the collinear condition, and a geometric calibration model is established based on this equation. A rotation transformation relationship between the image plane of the image space coordinate system and object plane of the local coordinate system is established by using the camera to observe the solar vector. Moreover, considering the angle readings of the pitch and azimuth encoders, centroid coordinates of the solar image and solar position parameters at different positions at different times, a multipoint observation equation is established, and the least squares method is used to solve the model. Geometric calibration of the equipment is conducted to determine the initial positions of the azimuth and pitch encoders. The mirror normal vector diagram is shown in Figure 2.

Assuming that the point light source is placed horizontally in the initial position, the pitch axis is orthogonal to the azimuth axis, and the central light axis of the reflector points to the north. This configuration is expressed as ${\left[\begin{array}{ccc} 0 & 1 & 0 \end{array}\right]}_{\mathrm{loc}}^{T}$ and ${\left[\begin{array}{ccc} 0 & 0 & 1 \end{array}\right]}_{\mathrm{mir}}^{T}$ in the northeast upper coordinate system and reflector coordinate system, respectively. At a certain moment, if the azimuth and altitude angles of the incident sunlight are $a_{\mathrm{azimuth}}$ and $a_{\mathrm{altitude}}$, respectively, the turntable rotates anticlockwise and clockwise around the pitch *X*-axis and azimuth axis, respectively. At this time, the central optical axis vector of the reflector coincides with the solar vector in the northeast upper coordinate system. In this case, in the local coordinate system, the transformation process from the optical axis vector of the mirror center to the coordinate rotation consistent with the solar vector can be expressed as
(1)$${\left[\begin{array}{c} X\\ Y\\ Z \end{array}\right]}_{\mathrm{loc}}=\left[\begin{array}{ccc} \cos(\alpha -\alpha_{0}) & \sin(\alpha -\alpha_{0}) & 0\\ -\sin(\alpha -\alpha_{0}) & \cos(\alpha -\alpha_{0}) & 0\\ 0 & 0 & 1 \end{array}\right]\left[\begin{array}{ccc} 1 & 0 & 0\\ 0 & \cos(\beta -\beta_{0}) & -\sin(\beta -\beta_{0})\\ 0 & \sin(\beta -\beta_{0}) & \cos(\beta -\beta_{0}) \end{array}\right]{\left[\begin{array}{c} 0\\ 1\\ 0 \end{array}\right]}_{\mathrm{loc}}$$
where α and β are the readings of the azimuth and elevation encoders at a certain time, respectively; $\alpha_{0}$ and $\beta_{0}$ are the initial position readings.

According to the definition of the coordinate system, if the mirror coordinate system is rotated anticlockwise by 90° around the axis, the local coordinate system coincides with the mirror coordinate system. According to the rotation matrix relationship of the coordinate transformation, the coordinate transformation relationship can be established at any point as follows:(2)$${\left[\begin{array}{c} X\\ Y\\ Z \end{array}\right]}_{\mathrm{loc}}=R_{X}^{-1}(\frac{\pi }{2}){\left[\begin{array}{c} x\\ y\\ z \end{array}\right]}_{\mathrm{mir}}.$$

Combining the coordinate rotation relation expressed in Equation (1) with the coordinate transformation and rotation relation expressed in Equation (2) yields
(3)$${\left[\begin{array}{c} X\\ Y\\ Z \end{array}\right]}_{\mathrm{loc}}=\left[\begin{array}{ccc} \cos(\alpha -\alpha_{0}) & \sin(\alpha -\alpha_{0}) & 0\\ -\sin(\alpha -\alpha_{0}) & \cos(\alpha -\alpha_{0}) & 0\\ 0 & 0 & 1 \end{array}\right]\left[\begin{array}{ccc} 1 & 0 & 0\\ 0 & \cos(\beta -\beta_{0}) & -\sin(\beta -\beta_{0})\\ 0 & \sin(\beta -\beta_{0}) & \cos(\beta -\beta_{0}) \end{array}\right]R_{X}^{-1}(\frac{\pi }{2}){\left[\begin{array}{c} x\\ y\\ z \end{array}\right]}_{\mathrm{mir}}.$$

In particular, when the optical axis vector of the mirror center is consistent with the solar vector, the coordinates of the solar vector in the mirror coordinate system are ${\left[\begin{array}{ccc} 0 & 0 & 1 \end{array}\right]}_{\mathrm{mir}}^{T}$, and the unit vector coordinates in the local coordinate system are ${\left[\begin{array}{ccc} X & Y & Z \end{array}\right]}_{\mathrm{loc}}^{T}$. According to Equation (3), the solar vector under the reflector can be transformed to the vector in the local coordinate system. Based on this aspect, the coordinate transformation relationship between the mirror and local coordinate systems is established based on the solar vector.

## 3. Geometric Calibration Modeling of the Turntable

### 3.1. Basic Calibration Model of the Turntable

In terms of the initial position of the point light source in the basic calibration model of the turntable, the *X*- and *Z*-axes in the mirror coordinate system coincide with the *E*- and *N*-axes in the local coordinate system, respectively. The central optical axis of the reflector points true north. The camera is affixed to the mirror assembly bracket, and the definition of its coordinate system is consistent with the mirror coordinate system. Therefore, the central optical axis vector of the reflector is replaced by the camera center optical axis vector. When the camera coordinate system is transformed to the local coordinate system, the relationship between the two coordinate systems must be established by multiplying the left side by the rotation matrix $R_{X}^{-1}(\frac{\pi }{2})$, as follows:(4)$${\left[\begin{array}{c} X\\ Y\\ Z \end{array}\right]}_{\mathrm{loc}}=R_{X}^{-1}(\frac{\pi }{2}){\left[\begin{array}{c} x-x_{0}\\ y-y_{0}\\ f \end{array}\right]}_{\mathrm{cam}}.$$

By combining Equations (3) and (4), the relationship between the camera and local coordinate systems can be established as
(5)$${\left[\begin{array}{c} X_{i}\\ Y_{i}\\ Z_{i} \end{array}\right]}_{\mathrm{loc}}=R_{Z}(\alpha_{i}-\alpha_{0})R_{X}(\beta_{i}-\beta_{0})R_{X}^{-1}(\frac{\pi }{2})\lambda \left[\begin{array}{c} x_{i}-x_{0}\\ y_{i}-y_{0}\\ f \end{array}\right]$$
where
$$R_{Z}(\alpha_{i}-\alpha_{0})=\left[\begin{array}{ccc} \cos(\alpha_{i}-\alpha_{0}) & \sin(\alpha_{i}-\alpha_{0}) & 0\\ -\sin(\alpha_{i}-\alpha_{0}) & \cos(\alpha_{i}-\alpha_{0}) & 0\\ 0 & 0 & 1 \end{array}\right],R_{X}(\beta_{i}-\beta_{0})=\left[\begin{array}{ccc} 1 & 0 & 0\\ 0 & \cos(\beta_{i}-\beta_{0}) & -\sin(\beta_{i}-\beta_{0})\\ 0 & \sin(\beta_{i}-\beta_{0}) & \cos(\beta_{i}-\beta_{0}) \end{array}\right].$$

$\alpha_{i}$ and $\beta_{i}$ are the azimuth and pitch encoder values corresponding to the encoder at a certain moment, respectively; $x_{i}$ and $y_{i}$ are the coordinates of the centroid of the solar image in the pixel coordinate system at a certain moment; and λ is the imaging scale factor. Moreover, $x_{0}$ and $y_{0}$ are the camera main point coordinates, and i represents the camera imaging time serial number or the solar position serial number at different times, with i=1⋯n.

We define $R_{\mathrm{cam}}^{\mathrm{loc}}=R_{Z}(\alpha_{i}-\alpha_{0})R_{X}(\beta_{i}-\beta_{0})R_{X}^{-1}(\frac{\pi }{2})$. Consequently, Equation (5) can be rewritten as
(6)$${\left(R_{\mathrm{cam}}^{\mathrm{loc}}\right)}^{-1}\left[\begin{array}{c} X_{i}\\ Y_{i}\\ Z_{i} \end{array}\right]=\lambda \left[\begin{array}{c} x_{i}-x_{0}\\ y_{i}-y_{0}\\ f \end{array}\right]$$
where $\left[\begin{array}{c} X_{i}\\ Y_{i}\\ Z_{i} \end{array}\right]=\left[\begin{array}{c} \sin\;a_{\mathrm{azimuth}}\cos\;a_{\mathrm{altitude}}\\ \cos\;a_{\mathrm{azimuth}}\cos\;a_{\mathrm{altitude}}\\ \sin\;a_{\mathrm{altitude}} \end{array}\right]$, $X_{i}$ represents the east (*E*) component of the sun in the local coordinate system, $Y_{i}$ represents the component of the sun due north (*N*) in the local coordinate system, and $Z_{i}$ represents the upward ($U_{p}$) component of the sun perpendicular to the earth plane in the local coordinate system.

Equation (6) represents the rotation transformation relationship between the image plane in the image space coordinate system and object plane in the local coordinate system. By dividing the first and second expressions of Equation (6) by the third expression, $x_{i}-x_{0}=a(n_{xi}-n_{x0})$ and $y_{i}-y_{0}=a(n_{yi}-n_{y0})$, where a is the pixel size and n is the number of pixels. Upon substituting this content into Equation (6), the basic calibration model of the turntable can be expressed as
(7)$$\left\{ \begin{array}{l} \frac{a}{f}(n_{xi}-n_{x0})=\frac{\cos(\alpha -\alpha_{0})X_{i}-\sin(\alpha -\alpha_{0})Y_{i}}{\sin(\alpha -\alpha_{0})\cos(\beta -\beta_{0})X_{i}+\cos(\alpha -\alpha_{0})\cos(\beta -\beta_{0})Y_{i}+\sin(\beta -\beta_{0})Z_{i}}\\ \frac{a}{f}(n_{yi}-n_{y0})=\frac{\sin(\alpha -\alpha_{0})\sin(\beta -\beta_{0})X_{i}+\cos(\alpha -\alpha_{0})\sin(\beta -\beta_{0})Y_{i}-\cos(\beta -\beta_{0})Z_{i}}{\sin(\alpha -\alpha_{0})\cos(\beta -\beta_{0})X_{i}+\cos(\alpha -\alpha_{0})\cos(\beta -\beta_{0})Y_{i}+\sin(\beta -\beta_{0})Z_{i}} \end{array}\right.$$

The right and left sides of the equation represent the calculation formula of the solar vector and optical axis vector of the turntable mirror center, respectively. When $x=x_{0}$ and $y=y_{0}$, the optical axis vector of the reflector points toward the sun. In this case, $a_{\mathrm{altitude}}=\beta -\beta_{0}$ and $a_{\mathrm{azimuth}}=\alpha -\alpha_{0}$. When $x\neq x_{0}$ and $y\neq y_{0}$, the optical axis vector of the reflector points toward a certain angle in space. In this case, $\theta_{\mathrm{altitude}}=\beta -\beta_{0}$ and $\varphi_{\mathrm{azimuth}}=\alpha -\alpha_{0}$.

In this manner, the relationship between the camera coordinate system and local coordinate system can be established by using the camera to observe the solar vector. Thus, any vector in the image space coordinate system can be transformed to the local coordinate system through the coordinate rotation transformation relationship. The solar vector observed by the camera represents the optical axis vector of the reflector. The control turntable uses the camera to realize data acquisition and automatic calibration in the local coordinate system.

### 3.2. High-Precision Geometric Calibration Model of the Turntable

The basic calibration model of the turntable is based on the assumption that the turntable is placed horizontally, the pitch axis is orthogonal to the azimuth axis, and the camera is positioned vertically. However, regardless of whether the actual turntable is horizontal, the pitch axis is vertical to the azimuth axis, and the camera is vertical. The levelness error, perpendicularity error, and camera placement perpendicularity error must be considered in the high-precision control system. In particular, to realize high-precision automatic calibration control of the turntable, it is necessary to establish a high-precision calibration model of the turntable and examine the geometric error parameters of the turntable obtained considering the basic calibration model. We consider that the error matrix of the turntable placement levelness is $R^{L}$, the orthogonal error matrix of the pitch and azimuth axes is $R^{S}$, and the vertical error matrix of the camera placement is $R^{C}$. In this case, the high-precision calibration model can be expressed as
(8)$${\left[\begin{array}{c} X_{i}\\ Y_{i}\\ Z_{i} \end{array}\right]}_{\mathrm{loc}}=\lambda R^{L}R_{Z}(\alpha_{i}-\alpha_{0})R^{S}R_{X}(\beta_{i}-\beta_{0})R^{C}R_{X}^{-1}(\frac{\pi }{2})\left[\begin{array}{c} x_{i}-x_{0}\\ y_{i}-y_{0}\\ f \end{array}\right]$$
where $R^{L}=R_{X}^{L}R_{Y}^{L}R_{Z}^{L}$, $R^{S}=R_{Z}^{S}R_{Y}^{S}R_{X}^{S}$, and $R^{C}=R_{X}^{C}R_{Y}^{C}R_{Z}^{C}$.

According to the rotation matrix, the same kind of rotation can be combined in the same direction. Equation (8) can be simplified to obtain a high-precision calibration model of the turntable as
(9)$${\left[\begin{array}{c} X_{i}\\ Y_{i}\\ Z_{i} \end{array}\right]}_{\mathrm{loc}}=\lambda R_{X}^{L}R_{Y}^{L}R_{Z}(\alpha_{i}-\alpha_{0})R_{Y}^{S}R_{X}(\beta_{i}-\beta_{0})R_{Y}^{C}R_{X}^{-1}(\frac{\pi }{2})\left[\begin{array}{c} x_{i}-x_{0}\\ y_{i}-y_{0}\\ f \end{array}\right]$$
where $R_{X}^{L}$, $R_{Y}^{L}$, and $R_{Z}^{L}$ represent the rotation matrix around the X, Y, and Z axes from the mirror coordinate system to the local coordinate system, respectively; $R_{Z}^{S}$, $R_{Y}^{S}$, and $R_{X}^{S}$ represent the rotation matrix around the Z, X, and Y axes from the pitch axis coordinate system to the azimuth axis coordinate system, respectively; and $R_{X}^{C}$, $R_{Y}^{C}$, and $R_{Z}^{C}$ represent the rotation matrix around the X, Y, and Z axes from the camera coordinate system to the mirror coordinate system, respectively. Consequently,
$$\begin{array}{c} R_{X}^{L}R_{Y}^{L}=\left[\begin{array}{ccc} 1 & 0 & 0\\ 0 & \cos\mu_{0} & -\sin\mu_{0}\\ 0 & \sin\mu_{0} & \cos\mu_{0} \end{array}\right]\left[\begin{array}{ccc} \cos\nu_{0} & 0 & \sin\nu_{0}\\ 0 & 1 & 0\\ -\sin\nu_{0} & 0 & \cos\nu_{0} \end{array}\right]R_{Y}^{S}=\left[\begin{array}{ccc} \cos\omega_{0} & 0 & \sin\omega_{0}\\ 0 & 1 & 0\\ -\sin\omega_{0} & 0 & \cos\omega_{0} \end{array}\right]\\ R_{Y}^{C}=\left[\begin{array}{ccc} \cos\gamma_{0} & 0 & \sin\gamma_{0}\\ 0 & 1 & 0\\ -\sin\gamma_{0} & 0 & \cos\gamma_{0} \end{array}\right] \end{array}$$
where $\mu_{0}$ and $\nu_{0}$ represent the level offset error of the turntable installation, $\omega_{0}$ represents the geometric error of the verticality of the pitch axis and azimuth axis of the turntable, and $\gamma_{0}$ represents the verticality offset error of the camera placement.

We define $R_{\mathrm{cam}}^{\mathrm{loc}}=R_{X}^{L}R_{Y}^{L}R_{Z}(\alpha_{i}-\alpha_{0})R_{Y}^{S}R_{X}(\beta_{i}-\beta_{0})R_{Y}^{C}R_{X}^{-1}(\frac{\pi }{2})=\left[\begin{array}{ccc} a_{1} & b_{1} & c_{1}\\ a_{2} & b_{2} & c_{2}\\ a_{3} & b_{3} & c_{3} \end{array}\right]$. By inserting Equation (9), we obtain
(10)$$\left\{ \begin{array}{l} \frac{a}{f}(n_{xi}-n_{x0})=\frac{X_{i}a_{1}+Y_{i}a_{2}+Z_{i}a_{3}}{X_{i}c_{1}+Y_{i}c_{2}+Z_{i}c_{3}}\\ \frac{a}{f}(n_{yi}-n_{y0})=\frac{X_{i}b_{1}+Y_{i}b_{2}+Z_{i}b_{3}}{X_{i}c_{1}+Y_{i}c_{2}+Z_{i}c_{3}} \end{array}\right..$$

Thus, a high-precision calibration model considering the geometric error of the system is established. However, in the process of automatic system calibration, camera lens distortion may produce errors, which may limit the increase in the calibration accuracy. Therefore, it is necessary to correct the lens distortion to further reduce the error sources. Considering the calibration model expressed in Equation (10), the chessboard calibration results are incorporated [27], and the lens distortion correction term is added. The first term approximation of the Taylor series expansion is adopted to correct the radial distortion error of the lens
(11)$$\left\{ \begin{array}{l} (x_{i}-x_{0})+\Delta x=f_{x}\frac{\bar{X}}{Z}\\ (y_{i}-y_{0})+\Delta y=f_{y}\frac{\bar{Y}}{Z} \end{array}\right.$$
where $x_{i}$ and $y_{i}$ are the coordinates of the image centroid in the pixel coordinate system; $x_{0}$ and $y_{0}$ are the camera main point coordinates; Δx and Δy are the radial distortion errors of the camera; $f_{x}$ and $f_{y}$ are the focal lengths of the camera in the x and y directions, respectively; and $\bar{X}=X_{i}a_{1}+Y_{i}a_{2}+Z_{i}a_{3}$, $\bar{Y}=X_{i}b_{1}+Y_{i}b_{2}+Z_{i}b_{3}$, and $Z=X_{i}c_{1}+Y_{i}c_{2}+Z_{i}c_{3}$.

According to the camera physical calibration model [28,29], the radial distortion error of the camera can be defined as follows:(12)$$\Delta x=\bar{x}k_{1}r^{2},\;\Delta y=\bar{y}k_{1}r^{2}$$
where $\bar{x}=(x_{i}-x_{0})$, $\bar{y}=(y_{i}-y_{0})$, and $r^{2}={\left(x_{i}-x_{0}\right)}^{2}+{\left(y_{i}-y_{0}\right)}^{2}$. Here, $k_{1}$ is the radial distortion coefficient of the camera, and r is the radial distance of the actual image point.

Substituting Equation (12) into Equation (11) yields a high-precision geometric error calibration model with camera distortion correction, as follows:(13)$$\left\{ \begin{array}{l} \frac{a}{f_{x}}(n_{x_{i}}-n_{x_{0}})\left\{1+a^{2}k_{1}\left[{\left(n_{x_{i}}-n_{x_{0}}\right)}^{2}+{\left(n_{y_{i}}-n_{y_{0}}\right)}^{2}\right]\right\}=\frac{\bar{X}}{Z}\\ \frac{a}{f_{y}}(n_{y_{i}}-n_{y_{0}})\left\{1+a^{2}k_{1}\left[{\left(n_{x_{i}}-n_{x_{0}}\right)}^{2}+{\left(n_{y_{i}}-n_{y_{0}}\right)}^{2}\right]\right\}=\frac{\bar{Y}}{Z} \end{array}\right..$$

Equation (13) represents the conversion of the solar vector in the local coordinate system to the representation in the image space coordinate system. Thus, the relationship between the solar vector observed by the camera in the image space coordinate system is established, and transformation from any vector in the image space system to the local coordinate system is realized. Finally, through actual camera observations, multipoint data are collected to establish multipoint observation equations to achieve high-precision calibration of the system installation geometric errors and verify the corresponding error parameters $\mu_{0}$, $\nu_{0}$, $\omega_{0}$, and $\gamma_{0}$, encoder initial positions $\alpha_{0}$ and $\beta_{0}$, and camera principal point and principal distance values $x_{0}$, $y_{0}$, $f_{x}$, and *f_y_*, among other factors. In this manner, high-precision calibration of the turntable system in the local coordinate system can be realized, leading to increased pointing accuracy.

## 4. Model Verification and Solution

### 4.1. Verification of the Model Coordinate Rotation Transformation Relationship

When the central light axis of the reflector points toward the sun, the coordinates of the solar vector in the mirror coordinate system are ${\left[\begin{array}{ccc} 0 & 0 & 1 \end{array}\right]}_{\mathrm{mir}}^{T}$, and the unit vector coordinates in the local coordinate system are ${\left[\begin{array}{ccc} X & Y & Z \end{array}\right]}_{\mathrm{loc}}^{T}$. First, forward verification is conducted according to Equation (3). By substituting ${\left[\begin{array}{ccc} 0 & 0 & 1 \end{array}\right]}_{\mathrm{mir}}^{T}$ and multiplying the three terms on the right side, we can obtain the vector representation of the sun in the local coordinate system, as follows:(14)$${\left[\begin{array}{c} X\\ Y\\ Z \end{array}\right]}_{\mathrm{loc}}=\left[\begin{array}{ccc} \cos(\alpha -\alpha_{0}) & \sin(\alpha -\alpha_{0})\sin(\beta -\beta_{0}) & \sin(\alpha -\alpha_{0})\cos(\beta -\beta_{0})\\ -\sin(\alpha -\alpha_{0}) & \cos(\alpha -\alpha_{0})\sin(\beta -\beta_{0}) & \cos(\alpha -\alpha_{0})\cos(\beta -\beta_{0})\\ 0 & -\cos(\beta -\beta_{0}) & \sin(\beta -\beta_{0}) \end{array}\right]{\left[\begin{array}{c} 0\\ 0\\ 1 \end{array}\right]}_{\mathrm{mir}}=\left[\begin{array}{c} \sin(\alpha -\alpha_{0})\cos(\beta -\beta_{0})\\ \cos(\alpha -\alpha_{0})\cos(\beta -\beta_{0})\\ \sin(\beta -\beta_{0}) \end{array}\right]$$
where $a_{\mathrm{azimuth}}=\alpha -\alpha_{0}$, and $a_{\mathrm{altitude}}=\beta -\beta_{0}$. The result is the same as that of the solar unit vector ${\left[\begin{array}{c} X\\ Y\\ Z \end{array}\right]}_{\mathrm{loc}}=\left[\begin{array}{c} \sin\;a_{\mathrm{azimuth}}\cos\;a_{\mathrm{altitude}}\\ \cos\;a_{\mathrm{azimuth}}\cos\;a_{\mathrm{altitude}}\\ \sin\;a_{\mathrm{altitude}} \end{array}\right]$ in the local coordinate system. Thus, the accuracy of the rotation matrix is preliminarily verified. Second, the vector representation of the sun in the local coordinate system is substituted into Equation (3) to calculate the vector representation of the sun in the mirror coordinate system, as follows:(15)$${\left(R_{\mathrm{cam}}^{\mathrm{loc}}\right)}^{-1}\left[\begin{array}{c} X_{\mathrm{loc}}\\ Y_{\mathrm{loc}}\\ Z_{\mathrm{loc}} \end{array}\right]=\left[\begin{array}{ccc} \cos(\alpha -\alpha_{0}) & -\sin(\alpha -\alpha_{0}) & 0\\ \sin(\alpha -\alpha_{0})\sin(\beta -\beta_{0}) & \cos(\alpha -\alpha_{0})\sin(\beta -\beta_{0}) & -\cos(\beta -\beta_{0})\\ \sin(\alpha -\alpha_{0})\cos(\beta -\beta_{0}) & \cos(\alpha -\alpha_{0})\cos(\beta -\beta_{0}) & \sin(\beta -\beta_{0}) \end{array}\right]\left[\begin{array}{c} \sin(\alpha -\alpha_{0})\cos(\beta -\beta_{0})\\ \cos(\alpha -\alpha_{0})\cos(\beta -\beta_{0})\\ \sin(\beta -\beta_{0}) \end{array}\right]=\left[\begin{array}{c} 0\\ 0\\ 1 \end{array}\right]$$

The calculation result for Equation (15) is the same as the vector representation ${\left[\begin{array}{ccc} 0 & 0 & 1 \end{array}\right]}_{\mathrm{mir}}^{T}$ of the sun in the mirror coordinate system when the optical axis of the reflector is aligned with the sun. Both the forward and reverse verification calculation results are the same as the predicted results, which demonstrates the accuracy of the coordinate rotation transformation matrix of the basic calibration model. The coordinate rotation transformation verification diagram for the calibration model is shown in Figure 3.

### 4.2. Model Solution

According to Equation (13), the geometric error parameters of the system to be calibrated are ($\mu_{0}$, $\nu_{0}$, $\omega_{0}$, and $\gamma_{0}$), the initial position parameters of the encoder are ($\alpha_{0}$ and $\beta_{0}$), and the camera parameters are ($x_{0}$, $y_{0}$, $f_{x}$, $f_{y}$, and *k*_1_). In total, 11 parameters exist. To solve the model, multipoint observations are needed. To this end, the multipoint observation equation is established, and the least squares method is used to solve the unknown parameters iteratively until the accuracy requirements are met. The solution process is as follows:(16)$$\begin{array}{l} w_{x}=\frac{a}{f_{x}}(n_{x_{i}}-n_{x_{0}})\left\{1+a^{2}k_{1}\left[{\left(n_{x_{i}}-n_{x_{0}}\right)}^{2}+{\left(n_{y_{i}}-n_{y_{0}}\right)}^{2}\right]\right\}-\frac{\bar{X}}{Z}\\ w_{y}=\frac{a}{f_{y}}(n_{y_{i}}-n_{y_{0}})\left\{1+a^{2}k_{1}\left[{\left(n_{x_{i}}-n_{x_{0}}\right)}^{2}+{\left(n_{y_{i}}-n_{y_{0}}\right)}^{2}\right]\right\}-\frac{\bar{Y}}{Z} \end{array}.$$

The first-order Taylor linearization expansion of Equation (16) is carried out at the initial value ${\left[\begin{array}{ccccccccccc} \mu_{0} & \nu_{0} & \omega_{0} & \gamma_{0} & x_{0} & y_{0} & f_{x} & f_{y} & k_{1} & \alpha_{0} & \beta_{0} \end{array}\right]}_{i}^{T}$, and the error equation is established:(17)$$\begin{array}{l} {w^{\prime }}_{x}=\frac{\partial w_{x}}{\partial \mu_{0}}\Delta \mu_{0}+\frac{\partial w_{x}}{\partial \nu_{0}}\Delta \nu_{0}+\frac{\partial w_{x}}{\partial \omega_{0}}\Delta \omega_{0}+\frac{\partial w_{x}}{\partial \gamma_{0}}\Delta \gamma_{0}+\frac{\partial w_{x}}{\partial x_{0}}\Delta x_{0}+\frac{\partial w_{x}}{\partial y_{0}}\Delta y_{0}\\ +\frac{\partial w_{x}}{\partial f_{x}}\Delta f_{x}+\frac{\partial w_{x}}{\partial f_{y}}\Delta f_{y}+\frac{\partial w_{x}}{\partial k_{1}}\Delta k_{1}+\frac{\partial w_{x}}{\partial \alpha_{0}}\Delta \alpha_{0}+\frac{\partial w_{x}}{\partial \beta_{0}}\Delta \beta_{0}\\ {w^{\prime }}_{y}=\frac{\partial w_{y}}{\partial \mu_{0}}\Delta \mu_{0}+\frac{\partial w_{y}}{\partial \nu_{0}}\Delta \nu_{0}+\frac{\partial w_{y}}{\partial \omega_{0}}\Delta \omega_{0}+\frac{\partial w_{y}}{\partial \gamma_{0}}\Delta \gamma_{0}+\frac{\partial w_{y}}{\partial x_{0}}\Delta x_{0}+\frac{\partial w_{y}}{\partial y_{0}}\Delta y_{0}\\ +\frac{\partial w_{y}}{\partial f_{x}}\Delta f_{x}+\frac{\partial w_{y}}{\partial f_{y}}\Delta f_{y}+\frac{\partial w_{y}}{\partial k_{1}}\Delta k_{1}+\frac{\partial w_{y}}{\partial \alpha_{0}}\Delta \alpha_{0}+\frac{\partial w_{y}}{\partial \beta_{0}}\Delta \beta_{0} \end{array}.$$

This equation is expressed in matrix form as
(18)$$\left[\begin{array}{c} {w^{\prime }}_{x}\\ {w^{\prime }}_{y} \end{array}\right]=\left[\begin{array}{ccc} \frac{\partial w_{x}}{\partial \mu_{0}} & \cdots & \frac{\partial w_{x}}{\partial \beta_{0}}\\ \frac{\partial w_{y}}{\partial \mu_{0}} & \cdots & \frac{\partial w_{y}}{\partial \beta_{0}} \end{array}\right]\left[\begin{array}{c} \Delta \mu_{0}\\ \vdots \\ \vdots \\ \Delta \beta_{0} \end{array}\right].$$

By using the camera multipoint observation, the multipoint observation equation is established as follows:$$L_{1}={\left[\begin{array}{c} {w^{\prime }}_{x,1}\\ {w^{\prime }}_{y,1} \end{array}\right]}^{0},L_{n}={\left[\begin{array}{c} {w^{\prime }}_{x,n}\\ {w^{\prime }}_{y,n} \end{array}\right]}^{0},\;x_{0}={\left[\begin{array}{c} \Delta \mu_{0}\\ \vdots \\ \vdots \\ \Delta \beta_{0} \end{array}\right]}^{i},\;A_{1}={\left[\begin{array}{ccc} \frac{\partial w_{x,1}}{\partial \mu_{0}} & \cdots & \frac{\partial w_{x,1}}{\partial \beta_{0}}\\ \frac{\partial w_{y,1}}{\partial \mu_{0}} & \cdots & \frac{\partial w_{y,1}}{\partial \beta_{0}} \end{array}\right]}^{0},\;A_{n}={\left[\begin{array}{ccc} \frac{\partial w_{x,n}}{\partial \mu_{0}} & \cdots & \frac{\partial w_{x,n}}{\partial \beta_{0}}\\ \frac{\partial w_{y,n}}{\partial \mu_{0}} & \cdots & \frac{\partial w_{y,n}}{\partial \beta_{0}} \end{array}\right]}^{0}.$$

We define
$$L=\left[\begin{array}{c} L_{1}\\ \vdots \\ L_{n} \end{array}\right],\;A=\left[\begin{array}{c} A_{1}\\ \vdots \\ A_{n} \end{array}\right],$$
where ${\left[\vdots \right]}^{0}$ represents the value at ${\left[\begin{array}{ccccccccccc} \mu_{0} & \nu_{0} & \omega_{0} & \gamma_{0} & x_{0} & y_{0} & f_{x} & f_{y} & k_{1} & \alpha_{0} & \beta_{0} \end{array}\right]}_{i}^{T}$. $L_{1}$ and $L_{n}$ denote the difference matrix between the solar vector observed by the camera and the calculated solar vector at the first and nth moment, respectively. In addition, ${w^{\prime }}_{x,1}$ and ${w^{\prime }}_{y,1}$ are the error components of the azimuth and pitch directions of the solar vector observed by the camera and the calculated solar vector at the first moment, respectively; ${w^{\prime }}_{x,n}$ and ${w^{\prime }}_{y,n}$ denote the error components of the azimuth and pitch directions of the solar vector observed by the camera and calculated solar vector at the nth moment, respectively; and $x_{0}$ is the matrix of the difference between the values of each variable and each corresponding expansion point. $A_{1}$ and $A_{n}$ denote the error equation at the first and nth moments, respectively, which are used to calculate the partial derivative matrix of each variable.

In this case, ***L*** = ***Ax***^0^, and we perform double left multiplication of $A^{T}$. After the positive definite treatment and matrix inversion, we obtain $x^{0}=(A^{T}A{)}^{-1}A^{T}L$. Subsequently, ***x***^0^ is substituted into the following expression to obtain the parameters to be solved:$$\begin{array}{l} {\left[\begin{array}{ccccccccccc} \mu_{0} & \nu_{0} & \omega_{0} & \gamma_{0} & x_{0} & y_{0} & f_{x} & f_{y} & k_{1} & \alpha_{0} & \beta_{0} \end{array}\right]}_{i+1}^{T}=\\ {\left[\begin{array}{ccccccccccc} \mu_{0} & \nu_{0} & \omega_{0} & \gamma_{0} & x_{0} & y_{0} & f_{x} & f_{y} & k_{1} & \alpha_{0} & \beta_{0} \end{array}\right]}_{i}^{T}+x^{0} \end{array}$$
where ${\left[\begin{array}{ccccccccccc} \mu_{0} & \nu_{0} & \omega_{0} & \gamma_{0} & x_{0} & y_{0} & f_{x} & f_{y} & k_{1} & \alpha_{0} & \beta_{0} \end{array}\right]}_{i}^{T}$ is the first-order Taylor expansion point value from the 0th to *i*th points (i=0⋯n).

Next, the Taylor expansion point is moved to the latest solution point ${\left[\begin{array}{ccccccccccc} \mu_{0} & \nu_{0} & \omega_{0} & \gamma_{0} & x_{0} & y_{0} & f_{x} & f_{y} & k_{1} & \alpha_{0} & \beta_{0} \end{array}\right]}_{i+1}^{T}$ expansion, and $x_{i+1}^{0}$ is solved again. The solution is iteratively found until the accuracy requirements are met.

### 4.3. Inverse Calculation of the Calibration Model

After solving the model, it is necessary to verify the results. After applying the calibration model, the encoder position coordinates *α* and *β* are calculated as the target value when the mirror normal vector and solar vector point in the same direction. Next, the servo motor is driven and controlled to move to the target position, and the camera collects the data for further verification. The model inverse solution algorithm after calibration is as follows.

According to the high-precision geometric calibration model, since the main point of the camera coincides with the image centroid coordinates when the mirror normal vector points toward the sun, that is, $x_{i}=x_{0}$ and $y_{i}=y_{0}$, the left term of the model is equal to zero. The right side of the model has a denominator $X_{i}c_{1}+Y_{i}c_{2}+Z_{i}c_{3}\neq 0$. Therefore, the following formula is established, and the inverse solution algorithm model can be expressed as
(19)$$\left\{ \begin{array}{l} X_{i}a_{1}+Y_{i}a_{2}+Z_{i}a_{3}=0\\ X_{i}b_{1}+Y_{i}b_{2}+Z_{i}b_{3}=0 \end{array}\right..$$

According to Equation (19), the azimuth and pitch α and β of the encoder, respectively, can be calculated by the least squares method when the normal of the reflector at different positions points toward the sun at different times. We define
$$\left\{ \begin{array}{l} {\bar{X}}_{re}=X_{i}a_{1}+Y_{i}a_{2}+Z_{i}a_{3}\\ {\bar{Y}}_{re}=X_{i}b_{1}+Y_{i}b_{2}+Z_{i}b_{3} \end{array}\right..$$

In this case, the α and *β* values satisfying the accuracy requirement can be determined using the following formula:(20)$$\underset{\alpha ,\;\beta }{\min}({\bar{X}}_{re}^{2}{+\bar{Y}}_{re}^{2}).$$

## 5. Experimental Results and Analysis

### 5.1. Reliability Analysis of Measured Data

Before obtaining the experimental data, the equipment is placed at the initial position, and the central light axis direction of the reflector is initially determined to be due north. To accelerate the calibration progress, reduce the calibration time, and test the encoder’s large-scale and multiple-angle motion characteristics, solar images at different positions of the camera array are collected. These images are used to perform the calibration model calculation and provide basic data to ensure accurate calibration. Using three techniques, three groups of data are collected to analyze the universality of the model solution. For the first group, the system moves from the right end to the left and collects two relatively irregular sets of pixel coordinate point data spread over the image plane of the detector. For the second group, the system moves from the right end to the left and collects a group of pixel coordinate points evenly distributed in the image plane of the detector. For the third group, the system moves from the left end to the right and collects a group of pixel coordinate points that are evenly distributed in the image plane of the detector. Moreover, the corresponding pitch, azimuth encoder readings and solar position parameters are recorded. The data acquisition path is shown in Figure 4.

Before the model is solved, the reliability of the experimental data is analyzed. The geometric parameters $\mu_{0}$, $\nu_{0}$, *ω*_0_, and $\gamma_{0}$ to be calibrated are set as 0, the calculated solar vector value of the three groups of data is considered the ordinate, the actual observation value of the optical axis vector of the mirror center is considered the abscissa for fitting analysis, and the calculated value of the solar vector is compared with the actual observation value. The comparison results are shown in Figure 5, Figure 6 and Figure 7, where $x_{i}-x_{0}$ and $y_{i}-y_{0}$ represent the actual solar vector pitch and azimuth components observed by the camera, respectively.

It can be seen from Figure 5, Figure 6 and Figure 7 that the data fitting results for the three groups of different paths indicate that the linear fitting correlation coefficient values between the calculated value of the solar vector and optical axis vector value of the mirror center observed by the camera are greater than 0.99. The linear fitting results are ideal, which further verifies the reliability of the experimental data and provides reliable basic data to solve the model.

### 5.2. Model Calculation and Theoretical Verification

The verified data solution model is used. The data of the model are shown in Table 1. Only 8 sets of data are listed in the table. The first row indicates the time of data collection. The second row indicates the corresponding pitch and azimuth encoder readings when the solar image is located at a certain position of the camera array. The third row indicates the altitude and azimuth of the sun in the local coordinate system corresponding to the data acquisition time.

In total, 105 sets of data are extracted from 221 sets of data to calculate the calibration model parameters. When the initial values of $\left[\begin{array}{cccccc} {µ}_{0} & \nu_{0} & \omega_{0} & \gamma_{0} & \alpha_{0} & \beta_{0} \end{array}\right]$, $\left[\begin{array}{ccc} x_{0} & y_{0} & k_{1} \end{array}\right]$, and $\left[\begin{array}{cc} f_{x} & f_{y} \end{array}\right]$ are [0 0 0 0 76 310] (unit:degree), [724 471 0.1063] pixels, and [15.6 mm 15.6 mm], respectively, the system parameters are [−0.1625 −0.178 0.10614 0.0345 77.19 310.49] (unit:degree), [719.03 470 −0.0009] pixels, and [15.614 mm 15.65 mm].

After the model is solved, it is necessary to evaluate the accuracy of the model parameters. First, the reliability of the results of the model is analyzed theoretically. The image centroid coordinates are used to represent the optical axis vector of the mirror as the X-axis, and the calculated solar vector value is considered the Y-axis in the fitting analysis. The linear fitting correlation coefficients of the two groups of values are considered to perform the reliability analysis of the evaluation model solution results. The fitting results of the two groups of data are shown in Figure 8.

It can be seen from the fitting results in Figure 8 that the image centroid coordinate represents the mirror normal direction consistent with the solar vector, and the fitting correlation coefficient $R^{2}$ is greater than 0.99998, which indicates a high linear correlation. Therefore, the reliability of the model results can be analyzed considering the theoretical data. Second, we analyze the error of the system calculation model. The system error caused by multipoint data optimization is used to analyze the pixel difference caused by the camera observation and angle difference caused by the encoder elevation and azimuth. The pixel, pitch, and azimuth error distributions corresponding to the systematic error distribution generated by the solution model are shown in Figure 9.

The error distribution data in Figure 9 show that the pixel error corresponds to the system model solution error, and the pixel average error and standard deviation in the X-axis direction are 1.253 pixels and 1.014 pixels, respectively. The average error and standard deviation in the Y-axis direction are 0.61 pixels and 0.45 pixels, respectively. The average error and standard deviation of the azimuth axis are 0.024° and 0.019°, respectively. The average error and standard deviation in the pitch axis direction are 0.012° and 0.0085°, respectively. According to the standard deviation data, these results are within the allowable error range. Therefore, from the theoretical error data, the reliability of the calculation model results is further verified.

### 5.3. Model Experiment Verification

In this step, we further verify the accuracy of the model parameters. Through the experiment, using the model inverse solution algorithm after calibration, the corresponding encoder pitch and azimuth target positions corresponding to the sun at different times are inversely solved, and the motor is driven to the target position. Finally, the accuracy of the model is verified by the actual observation of the camera. Part of the test data of the validated model is shown in Table 2, where only 8 sets of data are presented.

The first row in Table 2 indicates the solar altitude and azimuth angles when the central light axis of the reflector is aligned with the sun at different times. The second row indicates the target positions of the pitch and azimuth encoders, as calculated with the model inverse solution algorithm after calibration. The third row indicates the actual position measurement values of the encoder. The device considers the data presented in the second row as the target position, rotates the motor to the target position, and uses the encoder to detect the actual position as the feedback signal to further ensure the motion control accuracy of the turntable. The fourth row of data is the difference between the third row of data and the second row of data, which represents the pitch and azimuth control deviation. Figure 10 shows that the standard deviations of the pitch and azimuth angle control errors are 0.0176° and 0.0305°, respectively. The comparison and analysis of the pitch and azimuth encoder test data indicate that the model calculations are consistent with the measured values. The error range is approximately 0.04°, and the accuracy is better than 0.1°, which satisfies the verification requirements of the calibration model. The accuracy of the model is thus preliminarily verified by analyzing the motion control accuracy of the turntable and through actual observations by the solar observer.

Through the inverse calibration model, the motor is driven and controlled, and the model is preliminarily verified. To further verify the accuracy of the model parameters, by considering the actual observation of the camera after calibration, the solar image is tracked and collected, and the centroid coordinates of the solar image are used for verification. The centroid coordinates of the solar image at different times are compared with the camera main point coordinates to reflect the deviation degree of the center light axis of the reflector pointing toward the sun. The root mean square error (RMSE) of the two groups of data is calculated by Equation (21) to quantitatively evaluate the correctness of the model solving parameters and the tracking control accuracy of the system.

(21)$$\sigma (\theta )=\sqrt{\frac{\sum {\left(x-x_{0}\right)}^{2}}{n-1}}\;\sigma (\varphi )=\sqrt{\frac{\sum {\left(y-y_{0}\right)}^{2}}{n-1}}$$
Here, σ(θ) and σ(φ) are the RMSEs of the pitch and azimuth respectively; $x_{0}$ and $y_{0}$ are the coordinates of the principal point of the camera after calibration; x and y are the image centroid coordinates.

The centroid test data of the experimental verification model are presented in Table 3, where only 8 sets of data are listed.

The first row in Table 3 indicates the measured image centroid coordinates when the reflector centroid axis is aligned with the sun according to the target value of the inverse calibration model. The second row of data pertains to the use of a checkerboard to calibrate the camera’s main point coordinates. The third row shows the deviation between the measured image centroid and camera main point. The two sets of data and deviations are shown in Figure 11.

According to the two sets of data in Figure 9a,b, it can be determined by formula (21) that the RMSE values of the *X*- and *Y*-axis pixels are 2.0995 pixels and 0.8689 pixels, respectively, and the corresponding pixel angle resolution errors are 0.0377° and 0.0144°. The synthetic angular resolution error is calculated by formula (22) combined with the standard uncertainty formula [30], and the synthetic angular resolution error is 0.0403°.

(22)$$u_{c}=\sqrt{\sum_{i=1}^{N}{u_{i}}^{2}}$$
Here, $u_{i}$ is the component of error uncertainty.

It can be determined from the above analysis data that a small deviation exists between the centroid coordinates of the solar image obtained by the actual observation of the camera as the observation value and the coordinates of the main point of the camera as the real value. Nevertheless, the two sets of data are consistent, which demonstrates the accuracy of the calibration model. At the same time, the tracking control accuracy of the system is also measured through the RMSE. Because the tracking accuracy of the system represents the normal pointing accuracy of the mirror, the tracking control accuracy of the system is also the pointing accuracy of the system.

### 5.4. Accuracy Analysis of System Tracking

Through the experimental verification and analysis of the calibration model, the accuracy of system tracking using the model is evaluated. The tracking accuracy of the system mainly includes the motion control accuracy, external image processing algorithm accuracy and calibration model calculation accuracy. The accuracy of the motion control pertains to the accuracy (0.0003°) of the solar position calculated with the astronomical algorithm [31] and detection accuracy of the encoder (0.02°). The accuracy of the external image processing algorithm pertains to the accuracy of the image centroid extraction algorithm (0.032°) [32,33,34,35,36], average reprojection error of the camera calibration (0.1299 pixels), interference of the solar image noise and accuracy of the calibration model calculation. The uncertainty sources affecting the tracking accuracy of the system are presented in Table 4. The system tracking accuracy summarizes all the factors. The RMSE of the solar image obtained by the actual observation of the camera as the observation value and camera principal point coordinate as the real value is comprehensively evaluated as 0.0403°, and the tracking accuracy is noted to be better than 0.1°, which meets the requirements of the comprehensive pointing control accuracy of the system [37,38,39,40].

According to the data in Table 4, the uncertainty of the system calibration is approximately 0.0403°. That is, the tracking control accuracy of the system is 0.0403°, which is greatly improved compared with the tracking accuracy of the tracking equipment in the solar photovoltaic industry and the tracking accuracy of foreign point light sources [24,41,42,43,44,45,46]. This finding demonstrates the effectiveness of the calibration model in this paper.

Overall, the motion control error, encoder detection accuracy and image centroid extraction algorithm accuracy are the main error sources in the system control accuracy. Therefore, it is necessary to enhance the detection accuracy of the encoder, overcome the interference caused by the mechanical transmission error and unbalanced force in the motion processes, and optimize the image quality and image centroid extraction algorithm. Moreover, by enhancing the accuracy of the calibration camera and reducing the influence of the error caused by the model, the tracking accuracy of the system can be further increased to enhance the comprehensive pointing accuracy of the system and more effectively realize radiometric calibration and MTF detection of high-spatial resolution satellites.

## 6. Conclusions

A high-precision automatic geometric calibration modeling method for a point light turntable is proposed. Based on the principle of geometric calibration modeling, a high-precision automatic calibration model is established. By analyzing the reliability of the experimental data and solving the model, the feasibility and effectiveness of the method are demonstrated theoretically and experimentally. This approach can overcome the problem of the low precision of normal and single-point calibration, which limits the enhancement of the pointing accuracy. Moreover, the approach can reduce the calibration time, accelerate the calibration progress and increase the work efficiency, which facilitates high-frequency and high-efficiency networking automation to carry out the calibration of point light sources with different energy levels and increase the pointing accuracy of the system, achieve high-precision control of the central optical axis of the point light source reflector to point toward the target position, and reflect the light spot toward the satellite entrance pupil. Finally, this work lays a foundation for the high-precision, high-frequency, operational on-orbit radiometric calibration and MTF detection of high-resolution satellites. In addition, this system modeling method provides a theoretical basis for heliostat and solar photovoltaic equipment calibration.

## Figures and Tables

**Figure 1 sensors-21-02270-f001:**
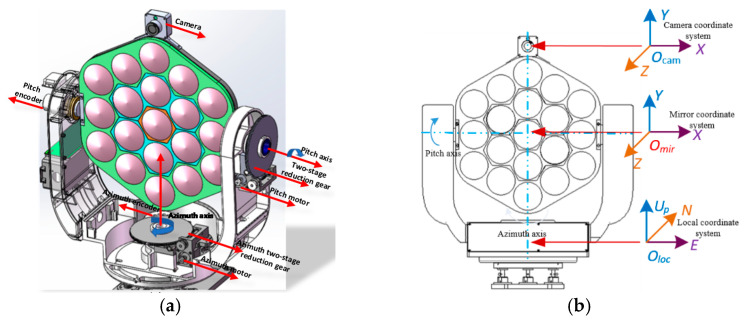
(**a**) Composition of the point light source system; (**b**) coordinate system establishment.

**Figure 2 sensors-21-02270-f002:**
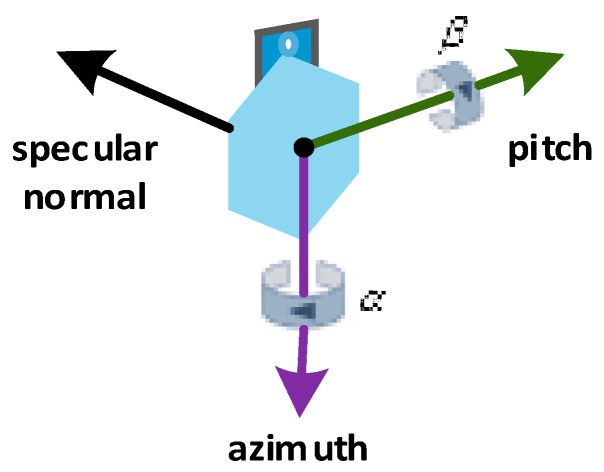
Mirror normal vector diagram.

**Figure 3 sensors-21-02270-f003:**
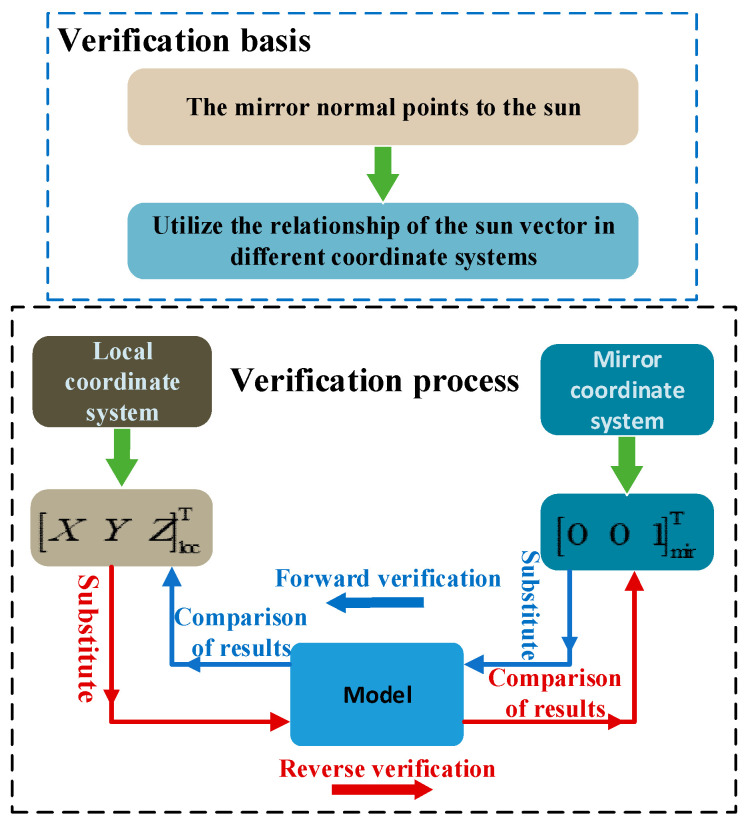
Coordinate rotation transformation verification of the calibration model.

**Figure 4 sensors-21-02270-f004:**
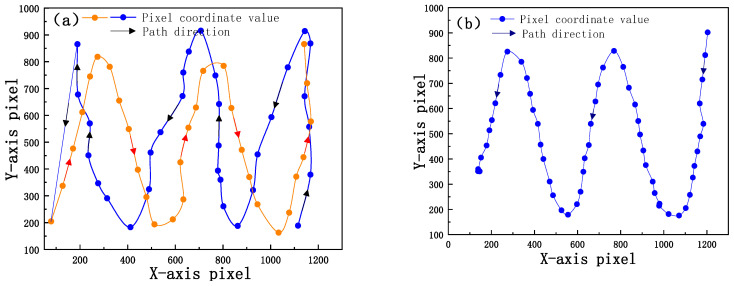
(**a**) First set of data; (**b**) second set of data; (**c**) third set of data.

**Figure 5 sensors-21-02270-f005:**
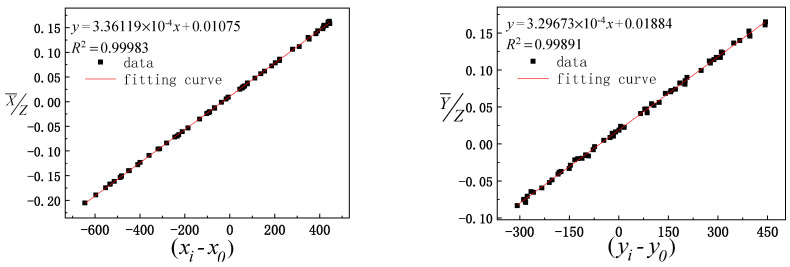
Fitting between the calculated solar vectors of the first group of data and actual observation values of the optical axis vector of the mirror center.

**Figure 6 sensors-21-02270-f006:**
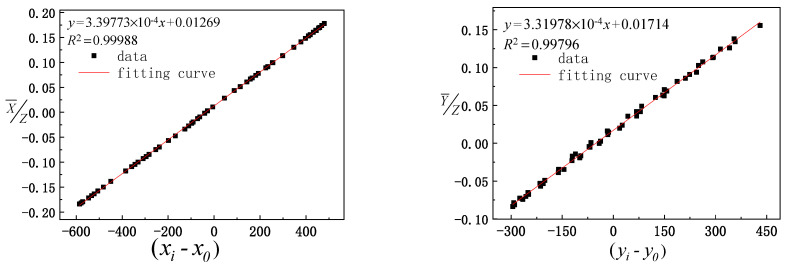
Fitting between the calculated values of the second group of data and actual observation values of the pointing mirror center optical axis.

**Figure 7 sensors-21-02270-f007:**
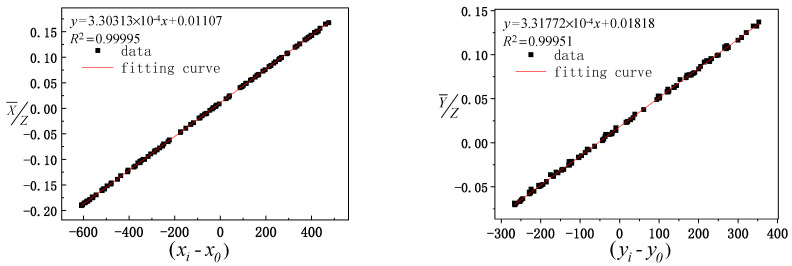
Fitting between the calculated values of the third group of data and actual observation values of the pointing mirror center optical axis.

**Figure 8 sensors-21-02270-f008:**
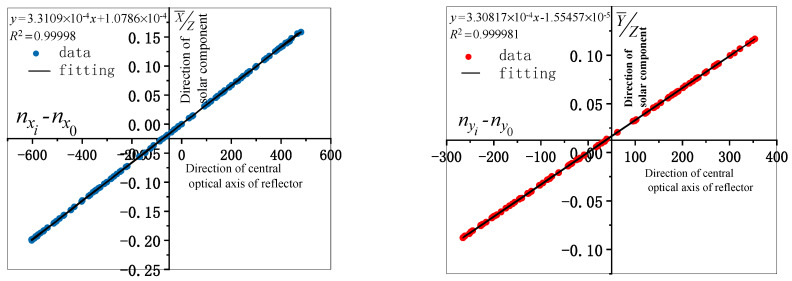
Fitting of the image centroid and solar vector.

**Figure 9 sensors-21-02270-f009:**
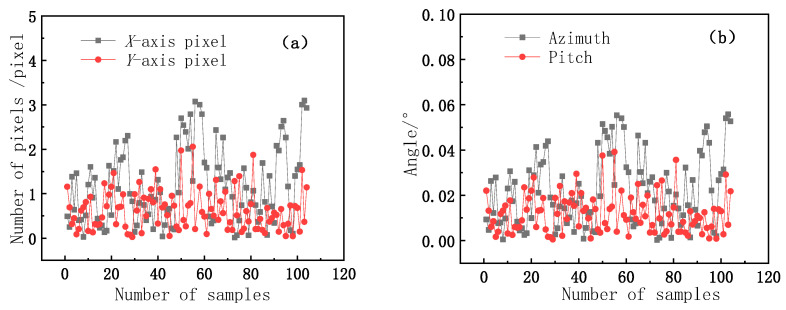
(**a**) Error distribution of the solution model corresponding to X and Y pixels; (**b**) pitch and azimuth angle error.

**Figure 10 sensors-21-02270-f010:**
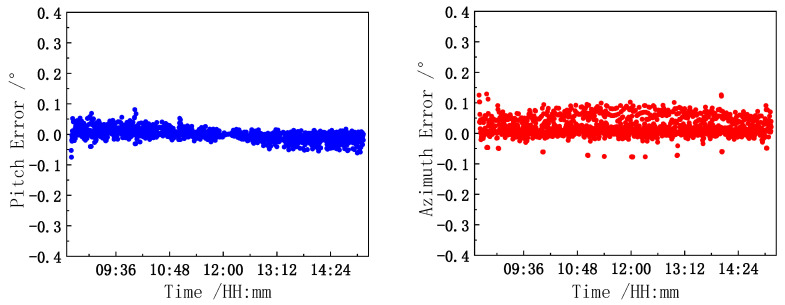
Pitch and azimuth control error.

**Figure 11 sensors-21-02270-f011:**
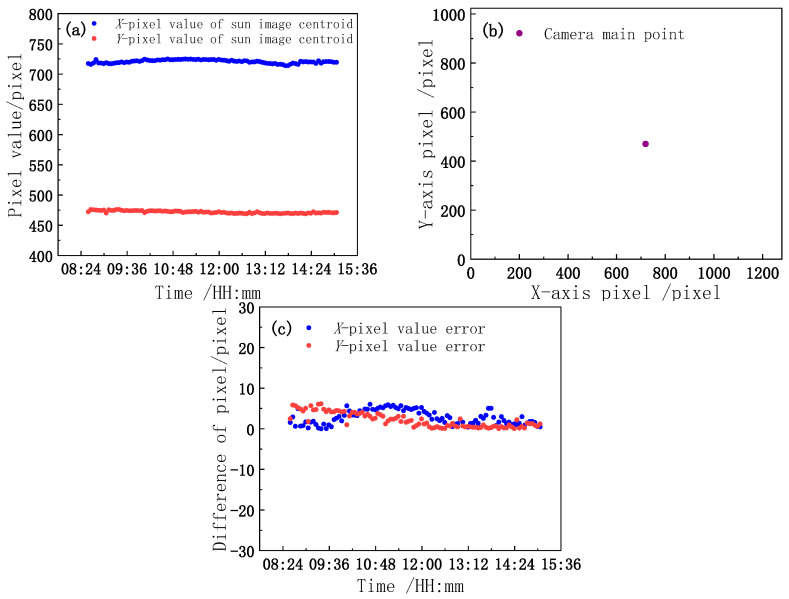
(**a**) Measured image centroid; (**b**) camera main point; (**c**) deviation.

**Table 1 sensors-21-02270-t001:** Data to solve the model.

Time	hh:mm:ss	9:06:53	9:09:09	9:11:46	9:14:33	9:17:14	9:21:06	9:24:12	9:27:23
Encoder angle value/(°)	Pitch	103.667	103.271	102.524	102.524	103.579	104.7	105.952	107.029
	Azimuth	167.563	167.256	167.278	167.256	167.585	167.278	166.663	166.355
Sun position/(°)	Altitude	29.041	29.397	29.804	30.234	30.644	31.227	31.688	32.156
	Azimuth	131.971	132.452	133.012	133.615	134.202	135.059	135.756	136.48
Sun centroid/(pixel)	Pixel x	206.383	216.259	244.145	273.59	317.017	345.299	351.167	372.528
	Pixel y	324.101	285.591	225.422	205.229	242.376	273.299	314.541	349.653

**Table 2 sensors-21-02270-t002:** Data used to solve the model.

Sun position/(°)	Altitude	29.811	32.342	34.575	37.129	39.697	39.58	37.73	30.35
	Azimuth	141.002	145.701	150.732	158.317	172.518	188.583	199.285	217.961
Encoder target angle of inverse solution/(°)	Pitch	106.787	109.314	111.555	114.126	116.719	116.653	114.829	107.534
	Azimuth	169.585	164.905	159.895	152.336	138.186	122.168	111.467	92.747
Encoder measurement angle/(°)	Pitch	106.831	109.292	111.577	114.17	116.741	116.697	114.807	107.512
	Azimuth	169.629	164.927	159.873	152.292	138.164	122.212	111.489	92.703
Error target and measurement/(°)	Pitch	0.044	−0.022	0.022	0.044	0.022	0.044	−0.022	−0.022
	Azimuth	0.044	0.022	−0.022	−0.044	−0.022	0.044	0.022	−0.044

**Table 3 sensors-21-02270-t003:** Partial centroid test data of the validated model.

Sun centroid/(pixel)	Pixel x	722.355	723.784	723.463	721.453	721.31	720.256	720.454	719.536
	Pixel y	474.071	472.398	473.061	469.46	472.441	469.79	471.393	470.585
Camera main point/(pixel)	Pixel x	719.000	719.000	719.000	719.000	719.000	719.000	719.000	719.000
	Pixel y	470.000	470.000	470.000	470.000	470.000	470.000	470.000	470.000
Error centroid and main point/(pixel)	Pixel x	3.355	4.784	4.463	2.453	2.31	1.256	1.454	0.536
	Pixel y	4.071	2.398	3.061	-0.54	2.441	-0.21	1.393	0.585

**Table 4 sensors-21-02270-t004:** Uncertainty analysis of system calibration.

Uncertainty Sources in System Tracking Control		Uncertainty and Error
Internal source	Motion control error	0.02°
	Astronomical algorithm accuracy	0.0003°
	Encoder detection accuracy	0.02°
External source	Image processing algorithm accuracy	0.030°
	Image centroid extraction algorithm accuracy	0.0204°
	Camera calibration average reprojection error	0.0021°
	Solar image noise interference	0.0219°
	Calibration model solution accuracy	0.180°
	Comprehensive evaluation accuracy	0.0403°

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
