# Peer review of "High-Precision Automatic Calibration Modeling of Point Light Source Tracking Systems"

_sensors, 2021, doi:10.3390/s21072270_

Round 1

Reviewer 1 Report

Dear authors,

I apologize, but I must say that I failed to understand how your work improved the current state of the art. As I still believe that some good effort is put in this work, I suggest major revision. Please see a few more comments below:

  1. You failed to explain well the current state of the art. It is not clear if someone previously designed and produced similar point light source tracking systems. It is not clear how your system is different from the existing ones. It is not clear in which characteristics it is better than state of the art solution. I would suggest also citing more diversified literature, as I can hardly believe that only research teams from south-east Asia are dealing with this topic.
  2. Many explanations are very ambiguous and hard to understand. The quantities that you present in the abstract are hard to interpret without giving some relation to the state-of-the-art solution quantities. It was not clear why exactly the tracking accuracy needs to be higher. It is not clear how much the tracking accuracy needs to be higher. Sentences in lines 31-33 ambiguous. Aggregating all references at the end of the paragraph (e.g. line 35) is ambiguous.
  3. It is common to have a short paragraph at the end of the introduction section explaining the article organization/structure if there is no table of content. Same goes at the beginning of each chapter that is subdivided on more subsections.
  4. Your figures are in most cases to small to see all of the details and in some cases in too low resolution. When you are explaining the content presented on the figures, it is common to make a reference to the figure early in the text. This helps readers to understand the description better.
  5. A lot of your sentences are short lists of the facts which sounds to some degree unnatural and it is hard to follow. I dislike when I get the similar comments on my articles, but I would suggest a correction from a native speaker.
  6. The amount of equations is extremely high, while the references to the sources of these equations are very sparse. Please consider removing some of the equations if they are considered as common knowledge or substituting some with the reference. I would also skip the validation of the equations by using different paths to formulate the same thing. I would say that empirical validation is sufficient to prove the point.
  7. Some of the used terms are a bit strange or misused: “groups of data” -> “usual is dataset”, in remote sensing reliability typically represents robustness towards outliers, I am not sure if it can be used the way you used it in text. Acceptable allowable error range (line 416) -> I presume one of the adjectives is redundant. And there are a few more instances of weird wording.

Reviewer 2 Report

The manuscript is about using the Sun and a set of mirrors for on-orbit radiometric calibration and spatial response characterization of satellite sensors. As the Sun is used as a point source of light that is directed toward the satellite by the mirrors, knowledge of the orientation of the mirrors is paramount for this method.

The manuscript describes how the orientation of the mirrors can be determined by using a camera that is mounted on the same frame as the mirrors. By tracking camera image of the Sun while changing orientation of the mirror frame, various parameters that characterize alignment of the system elements are calculated.

Not only a theoretical description of the optical model, but also results from actual tests of such a sun tracking system are presented. However, this is only the incident light side of the method, unless the mirrors are flat and perfectly aligned with the frame.

If the mirrors are convex, as suggested by Figure 1(a), the entire issue of what is the direction of the reflected light is not mentioned in the manuscript. While that is consistent with the manuscript’s title, it should be better explained in the Introduction and the Conclusions.

On line 387, six parameters are listed in the brackets, but only five numbers are shown on line 388, and only four on line 389.

Reviewer 3 Report

This is an interesting paper that describes a novel method for geometric calibration of satellite sensors using mirror arrays on the ground.  It should be of interest to the remote sensing community and therefore the paper is acceptable for publication after addressing the following issue:

The paper focuses on geometric calibration, but made a claim about "radiometric" calibration without further discussion.  It's not clear how this can be used for "radiometric" calibration so I would suggest the authors to address this issue in the revision.

Round 2

Reviewer 1 Report

Dear authors,

Thank you for your nice and detailed answers to my questions. Now I completely understood the contribution of your work and I think it is significant and worth publishing.